# Docetaxel Administration via Novel Hierarchical Nanoparticle Reduces Proinflammatory Cytokine Levels in Prostate Cancer Cells

**DOI:** 10.3390/cancers17111758

**Published:** 2025-05-23

**Authors:** Ravikumar Aalinkeel, Satish Sharma, Supriya D. Mahajan, Paras N. Prasad, Stanley A. Schwartz

**Affiliations:** 1VA Western New York Healthcare System, Buffalo, NY 14215, USA; 2Department of Urology, University at Buffalo, Buffalo, NY 14203, USA; ss466@buffalo.edu; 3Division of Allergy, Immunology and Rheumatology, Department of Medicine, University at Buffalo, Buffalo, NY 14203, USA; smahajan@buffalo.edu; 4Department of Chemistry, and The Institute for Lasers, Photonics, and Biophotonics, University at Buffalo, Buffalo, NY 14260, USA; pnprasad@buffalo.edu

**Keywords:** nanoparticle, hierarchical nanoparticle, inflammation, cytokines, Castration Resistant Prostate Cancer, docetaxel resistance

## Abstract

Docetaxel (Doc) resistance in prostate cancer (CaP) is linked to the secretion of proinflammatory cytokines, which facilitate interactions between tumor cells and macrophages, contributing to treatment resistance. This study developed a Doc-resistant CaP cell line (LNCaP-Doc/R) to investigate whether encapsulating Doc in a PLGA-Chitosan core-shell hierarchical nanoparticle (HNP-Doc) could reduce proinflammatory signaling. Results showed that LNCaP-Doc/R cells required significantly higher Doc concentrations for effective treatment. Cells treated with free Doc exhibited markedly higher levels of proinflammatory cytokines compared to those treated with HNP-Doc. Additionally, co-culturing LNCaP-Doc/R cells with monocytes revealed reduced macrophage differentiation markers when treated with HNP-Doc. These findings suggest that nanoparticle-encapsulated Doc can mitigate the proinflammatory response associated with resistance, potentially improving therapeutic outcomes in CaP treatment.

## 1. Introduction

The most common non-cutaneous cancer in American men is prostate cancer (CaP), which is also the second leading cause of cancer mortality in men in the US [1]. In 2024, ~299,490 males were diagnosed with CaP, and ~35,500 died from it [1]. The primary treatment for advanced and metastatic CaP is androgen deprivation therapy (ADT) [2]. Although CaP is initially susceptible to ADT, eventually resistance to ADT occurs, and CaP becomes hormone-refractory and typically metastasizes to the bone [3,4]. Notably, half of the patients treated with ADT do not respond to this treatment [5,6]. Failure to respond to ADT results in Castration-Resistant Prostate Cancer (CRPC), which lacks effective therapies [3,4]. The conventional drug of choice for the management of CRPC is docetaxel (Doc), which shows overall survival benefits of ~3 months [7,8,9]. However, resistance to Doc management also sets in, and further treatment for CRPC is only palliative. CRPC treatments often fail due to intrinsic/acquired resistance and/or dose-limiting adverse effects. Therapy resistance pathways include overexpression of P-glycoprotein [10], increased STAT1 expression, and inflammatory cytokines [11].

Considerable advancements have occurred in nanomedicine and drug delivery over the past few decades, leading to the emergence of a variety of nanoparticles (NPs) applicable in medical and pharmacological contexts, especially for potential use in cancer treatment, infectious diseases, and neurological disorders [12,13]. NPs have been modified to be compatible with living organisms, causing minimal or no harm to healthy cells and tissues. Additionally, they are designed to transport sizable amounts of therapeutic drugs and gene therapy agents [14,15]. Moreover, precise targeting can be achieved by chemically conjugating aptamers, specific peptide ligands, or monoclonal antibodies to the surface of NPs, leading to site-directed therapy [16,17]. Enclosing chemotherapeutic agents within NPs and directing them to specific cellular targets mitigates systemic toxicity and potentially reduces the risk of proinflammatory response [18]. Additionally, encapsulating highly labile gene therapy agents such as antisense nucleotides or silencing RNAs(siRNAs) in NPs establishes a secure environment for the payload during systemic administration [19,20]. A novel class of NPs, termed “Theranostic NPs”, has been developed to integrate therapeutic and imaging capabilities, facilitating real-time, image-guided monitoring of the distribution of their therapeutic payloads [21,22]. Recent advancements in nanotechnology have been applied to the treatment of advanced-stage prostate cancer, offering a promising opportunity for a previously unfavorable prognosis [23,24].

Over the years, we have developed novel NPs for the therapy of CaP. Our earlier studies concentrated on therapeutic cationic polylactide (CPLA) designed for the delivery of IL-8 siRNA in the treatment of CaP [25]. Subsequently, our unique CPLA-based nano-systems, enabling both drug(doxorubicin) and gene delivery (IL-8 siRNA), were designed [26]. Although this was a groundbreaking approach, it suffered from the limitation of using a chemotherapeutic agent, doxorubicin (dox), which was not the frontline drug of choice for CRPC treatment [8,9,27]. Subsequently, we have made a quantum leap and developed the next generation of theranostic, multifunctional, hierarchical NPs (HNPs) built with FDA-approved materials for the simultaneous delivery of drug and imaging agents for clinical applications [28]. Our multifunctional hierarchical nanoparticle (HNP) consists of biodegradable chitosan (CS) wrapped Poly (lactic-co-glycolic acid) (PLGA) NPs loaded with CRPC-approved drug Doc and up-conversion nanoparticles (UCNPs), with the latter coated with IL-8 siRNA for gene silencing. The IL-8 siRNA incorporated as part of our HNP is to silence IL-8 gene expression, rendering tumor cells less likely to undergo metastasis and/or angiogenesis [20]. The combination of IL-8 siRNA, UCNPs, and encapsulated Doc demonstrated a dramatic ~14,000-fold increase in efficacy in killing PC-3 CaP cells over free-Doc, at the same time providing dual-modality imaging capabilities that facilitate image-guided, real-time monitoring of therapy for a truly theranostic platform to treat CaP [28]. Our nano-formulation (NF) dramatically enhanced therapeutic actions at significantly lower concentrations and can potentially reduce or eliminate toxicity to healthy bystander cells and tissues in future applications.

Recent studies aimed at elucidating the mechanisms of treatment resistance in cancers, conducted both in vitro and in vivo, have identified the proinflammatory response by tumor cells as a crucial factor [29,30]. These studies demonstrated a positive correlation between cytokines related to macrophage recruitment and activation and chemotherapy resistance, both in vitro and in vivo. In a separate clinical study, individuals who poorly responded to chemotherapy exhibited increased levels of IL-1ra, a pro-inflammatory cytokine [31]. Similarly, Wu et al. have shown that the levels of MIC1, a cytokine associated with macrophage functioning, increased in the serum of poor responders to chemotherapy and in vitro in cells treated with Doc [32]. The findings from these in vitro and clinical studies signal a role for macrophages and pro-inflammatory cytokines in the emergence of Doc resistance in CRPC. Tumor cell-derived cytokines released in response to increased intracellular concentrations of Doc are responsible for attracting monocytes and macrophages to the tumor site to induce Doc resistance [29,33]. This is a great concern for continued treatment with Doc despite the drug showing clear benefits in overall survival rates [34,35]. This proinflammatory response could potentially be overcome by administering Doc encapsulated in our HNP, theoretically protecting “Doc” from exposure to the immune system and mellowing down the cytokine release. This present study aimed to determine if our HNP-based Doc delivery reproduces the same effect in a different metastatic CaP cell line, LNCaP, as we observed in our previous study with the CaP cell line PC-3 [28]. Further, we also wanted to test if an HNP-Doc administration mellows down cytokine release associated with Doc resistance in vitro, followed by studies to understand the interaction between macrophages and CaP cells using U937 cells, a promonocytic cell line, that can be induced to terminal macrophage differentiation and examined the difference in cytokine production between LNCaP or docetaxel resistant LNCaP (LNCaP-Doc/R) cells, with or without U937 co-culture, to assess the interaction between the macrophages and the CaP cells.

## 2. Methods

**Nanoparticle Synthesis and Characterization:** HNPs were synthesized via a nano-emulsion method and characterized by dynamic light scattering (DLS), zeta potential analysis, transmission electron microscopy (TEM) was performed with JEM-2010 microscope (JEOL USA, Inc., Peabody, MA, USA),UV–Vis spectroscopy, and high-performance liquid chromatography (HPLC) from, Waters Corporation (Milford, MA, USA). Drug release was evaluated via dialysis in acetate buffer (pH 5.5) and phosphate-buffered saline (PBS, pH 7.4) over 5 days, as already published elsewhere [28].

**Cells and Culture Conditions:** LNCaP, PC-3, and U937 cell lines were sourced from the American Type Culture Collection (Manassas, VA, USA). LNCaP exhibits significantly lower aggressive phenotypes, as demonstrated by us, are positive for PSMA expression, responsive but not growth dependent upon androgen, whereas PC-3 are highly aggressive, fast-growing, androgen-independent CaP cells [36]. Cells were cultured at 37 °C in a humidified environment comprising 95% air and 5% CO_2_. They were sustained in RPMI-1640 medium enriched with non-essential amino acids, L-glutamine, a twofold vitamin solution (Life Technologies, Grand Island, NY, USA), sodium pyruvate, Earle’s balanced salt solution, 10% fetal bovine serum, and penicillin and streptomycin (Flow Labs, Rockville, MD, USA).

**Development of Docetaxel Resistant LNCaP (LNCaP-Doc/R) Cell Line:** In consideration of the absence of consensus in the literature related to the development of Doc-resistance, we employed an approach demonstrated to be efficient in our laboratory. This method entails the progressive exposure of LNCaP cells to increasing concentrations of Doc to elicit resistance in LNCaP cells. The CaP cells were treated with an initial dosage of Doc (0.1 nM) for 24 h. After 24 h, the dead cells were eliminated. The surviving cells were collected and replated and allowed to recover by culturing in RPMI-1640 medium enriched with non-essential amino acids, L-glutamine, a twofold vitamin solution (Life Technologies, Grand Island, NY, USA), sodium pyruvate, Earle’s balanced salt solution, 10% fetal bovine serum, and penicillin and streptomycin (Flow Labs, Rockville, MD, USA) for three cycles of one week each and exposed to a higher dosage of 0.2 nM. The procedure was reiterated for doses of 0.5, 1.0, 2.0, 5.0, 10.0, and 25 nM of Doc. The cells demonstrating resistance to 25 nM of Doc were harvested and labeled as LNCaP-Doc/R. The cell conditions were meticulously observed during the gradual escalation of Doc concentration. The elevation of Doc concentration had to be delayed in the presence of several dead cells or when cell cloning proceeded at a sluggish rate. U937 cells underwent differentiation through treatment with 100 ng ml^−1^ phorbol 12-myristate 13-acetate (PMA; Sigma, St Louis, MO, USA) for 24 h, followed by a recovery period of 96 h before Doc treatment. For assays involving conditioned media from cell lines, cells were grown for 5 days before substituting the culture media with fresh media, with or without the inclusion of Doc (5 ng/mL). The conditioned media were harvested after 24 h, centrifuged at 1500× *g* for 5 min, and the supernatant was preserved at −80 °C for further quantification. In co-culture experiments, LNCaP and LNCaP-Doc/R cells were simultaneously seeded in the same flask with undifferentiated U937 cells. The media was changed after 24 h to eliminate non-adherent cells. Cells received Doc treatment at a concentration of 5 ng/mL for 24 h on day 3, and the supernatant was collected in 24 h.

**Determination of Growth Curves and Resistance Index for LNCaP and LNCaP-Doc/R cells:** Cell proliferation was assessed using the Cell Counting Kit-8 (CCK-8; cat. no. 96992, Sigma-Aldrich, St. Louis, MO, USA). LNCaP and LNCaP-Doc/R cells, cells were seeded into 96-well plates at a density of 1.0 × 10^4^ cells per well. Following an incubation for the required amount of time at 37 °C in a 95% humidified atmosphere with 5% CO_2_, 10 µL of CCK-8 solution was added to each well and incubated for an additional hour at 37 °C. Absorbance at 450 nm was then measured using a microplate spectrophotometer (BioTek Instruments, Inc., Winooski, VT, USA). A standard curve was constructed with different numbers of cells and OD, and the OD of the unknown was compared against the standard curve to determine the number of cells at various time points. This measurement was repeated independently every day until day 5, and growth curves were generated. Each experiment was conducted three times, and the average values were used. The cell doubling time was calculated using the following equation Doubling Time(Dt) = [T × (ln2)]/[ln(Xe/Xb)], where T is time in any units, Xb is the starting number of cells, Xe is the final number of cells and ln2 is constant. The resistance index (RI) of Doc in LNCaP-Doc/R cells was determined by calculating the ratio of the IC_50_ of the resistant cell line to that of the sensitive cell line.

**Cytotoxicity Assay for Proliferation Inhibition and IC_50_ quantification for Docetaxel in Naïve and Doc-Resistant Cells**: To assess the cytotoxicity of docetaxel (Doc) and resistance to Doc, the MTT assay was employed. LNCaP and LNCaP-Doc/R cells were seeded in 96-well plates at a density of 1 × 10^4^ cells per well and incubated at 37 °C in a humidified atmosphere with 5% CO_2_ for 24 h. Following incubation, the cells were washed with PBS and exposed to a range of Doc concentrations in cell growth media, starting at 120 µM, in a stepwise 10-fold serial dilution. After a 48 h incubation period, 10 µL of MTT solution was added to each well and incubated for 4 h to allow for formazan formation. Formazan crystals were then dissolved using the provided solvent, and 100 µL of the resultant solution was transferred to each well. Absorbance at 570 nm was measured using a microplate spectrophotometer (BioTek Instruments, Inc., Winooski, VT, USA). All experiments were conducted in triplicate, and average values were used for analysis. Control values were set at 0% cytotoxicity or 100% cell viability. Cytotoxicity data were fitted to a sigmoidal curve using GraphPad Prism 10.0 software (San Diego, CA, USA), and a three-parameter logistic model was used to calculate the inhibitory concentration-50 (IC_50_). Resistance indices (RIs) were calculated by comparing the 50% inhibitory concentration (IC_50_) values of LNCaP-Doc/R cells to those of LNCaP cells.

**RNA Extraction and Real-Time Quantitative Polymerase Chain Reaction (RT-PCR) for Docetaxel-Resistant Gene Expression:** Cytoplasmic RNA was extracted using the acid guanidinium thiocyanate–phenol–chloroform method, as described previously [37], utilizing TRIzol^®^ reagent (Invitrogen, Waltham, MA, USA). The final RNA pellet was dried, resuspended in diethyl pyrocarbonate-treated water, and its concentration was determined via spectrophotometry at 260 nm. The relative expression of ABCB1 (P-glycoprotein), cadherin and vimentin, NOTCH2, HES1, and RICTOR, was assessed using the SYBR Green master mix from Stratagene (La Jolla, CA, USA) to perform quantitative PCR (qPCR) with the Stratagene MX3005B system (La Jolla, CA, USA). Threshold cycle differences were employed to quantify the relative abundance of PCR targets within each sample tube [38]. Relative mRNA expression levels were calculated as the transcript accumulation index (TAI = 2^ΔΔCT^), using the comparative CT method [39]. To ensure accuracy, all data were normalized based on RNA input quantity by measurements performed on the reference gene β-actin. Additionally, results from RNA-treated samples were normalized to those obtained from RNA of untreated control samples.

**Imaging studies:** After incubation with Nile Red-labeled HNP for 72 h, cells are washed three times with PBS and then fixed with 4% paraformaldehyde at RT for 30 min. Cells are then counter-stained with DAPI (nuclear stain) and imaged using the EVOS^®^ FL Cell Imaging System (Life Technologies, Grand Island, NY, USA), and the Intracellular fluorescence is quantified using Image J. version 1.54f (National Institutes of Health, Bethesda, MD, USA) (https://imagej.net/ij/ (accessed on 21 October 2024)). Untreated cells were used as a negative control.

**Doc Quantification by HPLC:** To 100 µL of cell lysate, 300 µL of acetonitrile: methanol (1:1) containing internal standard (0.1 µg/mL), and ascorbic acid (0.5 mg/mL) is added to precipitate proteins. The suspension is centrifuged at 18,000 RCF for 15 min, and 2 µL is injected into an HPLC column system for quantification of the drug. The mobile phase is a mixture of acetonitrile and 0.1% TFA (50:50, *v*/*v*, HPLC grade, Fisher Scientific, Pittsburgh, PA, USA), filtered through a 0.25 μm pore membrane filter, and eluted at a flow rate of 1 mL/min. Effluents will be monitored at 232 nm. Calibration standards and quality control samples were included with each analytical run as per standard operating procedures.

**Analysis of cytokines in vitro:** Cytokines in culture supernatant were measured by ELISA kits from R and D Systems, Minneapolis, MN, USA according to the manufacturer’s instructions. Cytokines quantified include C5/C5a, CD40L, GCSF, GMCSF, GROα, I-309, IFNγ, IL-1α, IL-1ra, IL-1β, IL-2, IL-4, IL-5, IL-6, IL-8, IL-10, IL-12 IL-13, IL-16, IL-17, IL-17E, IL-23, IL-27, IL-32α, MCP1, MIF, Serpin E1, RANTES.

**Apoptosis Assay:** The apoptosis assay was conducted by seeding cells at a density of 10 × 10^3^ cells per 250 μL of medium per 0.95 cm^2^ growth area into 48-well plates. Following overnight incubation, the cells were treated with either free-Doc or HNP-Doc for 24 h. To ensure consistency, the concentrations of the drug solvents in all wells, including the control wells, were adjusted to match the highest concentration of a given solvent used. After the treatments, the assay was performed by the protocol outlined in the FITC Annexin V/Dead Cell Apoptosis Kit using FITC Annexin V and propidium iodide (PI) reagents (Invitrogen™, Grand Island, NY, USA). We used RWD C100-Pro automated cell counter (Sugar Land, TX, USA), which utilizes dual-fluorescence channel optical design and high-performance optical components to capture and analyze fluorescence signals and quantify viable and dead cells based on fluorophore. The C100 incorporates intelligent image recognition technology to accurately identify and count fluorescent cells for cell counting, employing % dead cells vs live and calculating the ratio. For each well, the ratio of apoptotic cells to the total number of cells was determined based on cell counter output. Each experimental condition was prepared and analyzed twice in triplicate.

**Statistical Analysis:** All experiments were repeated at least four times in triplicate. Values are expressed as mean ± SD fold change. The significance of the difference between the control and each experimental test was analyzed by an unpaired *t*-test (GraphPad, Prism 10.0, Boston, MA, USA), and a value of *p* < 0.05 was considered statistically significant.

## 3. Results

**Development of the Docetaxel-Resistant LNCaP/Doc-R Cell Line and Further Characterization:** The LNCaP-Doc/R cell line was established by intermittently exposing LNCaP cells to increasing concentrations of Doc over 8 months, as detailed in the methods section. Cellular phenotypes were documented using an inverted microscope at 10X and 20X magnifications (Figure 1A, top and bottom panels, respectively). Our images indicate that compared to the LNCaP cells, the LNCaP-Doc/R cells appeared small and had more irregular cell margins. Further, cell growth characteristics were evaluated using the Cell Counting Kit-8 (Sigma-Aldrich, St. Louis, MO, USA), and doubling times (Dt) were calculated following optical density (OD) measurements. Dt for LNCaP cells and LNCaP-Doc/R cells were 58.05 and 56.42 h, respectively (Figure 1B). When the dose effect for Doc was evaluated for cell survival in proliferation assays, we observed a dose-dependent effect in both the cell lines, and the determined IC_50_ values were 10.1 ± 0.1 pM for LNCaP cells and 918.2 ± 10.04 pM for LNCaP-Doc/R cells, respectively (Figure 1C). The resistance index of LNCaP-Doc/R cells was ~90-fold higher than that of LNCaP cells, confirming significant Doc-resistance.

**Doc-Resistance Confirmation by Quantifying Markers of Doc-Resistance by QPCR:** Resistance of tumor cells to Doc treatment could also be the result of changes in the expression of various genes related to drug transport, EMT pathway, mTOR2 pathways, and cytokine secretion. To understand if one or all pathways are involved, we performed gene expression analysis on a panel of genes in these pathways in LNCaP-Doc/R cells and naïve LNCaP cells. The selected genes are known for their involvement in chemotherapy resistance pathways, like drug transporters (ABCB1 (P-glycoprotein), EMT pathway (cadherin and vimentin), Notch and Hedgehog signaling pathway (NOTCH2, HES1), and mTORC2 signaling pathway (RICTOR). RNA was reverse-transcribed, and cDNA amplified by qPCR using primers specific for the genes of interest and the housekeeping gene, β-actin. Experiments were repeated three times, and results are shown in Figure 1D. Our results show LNCaP-Doc/R cells significantly upregulated the gene expression of ABCB1 p-glycoprotein ~10-fold (Figure 1D, *p* < 0.001), EMT pathway genes cadherin ~ 8-fold and vimentin ~3-fold, the NOTCH2 genes ~2.8-fold, Hedgehog signaling genes ~2.7-fold, and mTORC2 signaling pathway genes by ~ 3.2-fold.

**Nanoparticle Synthesis and Characterization of our HNP:** Fabrication and HNPs containing PLGA, CS, Doc, and UCNPs were synthesized using a water/oil/water emulsion method and published elsewhere [28]. Doc incorporation was achieved through the addition of Doc to the methylene chloride (MC) phase, while the incorporation of UCNPs was facilitated by their addition to the initial water phase. The size of the HNPs and the confirmation of the retention of positive charge have also been documented in our previous study.

**Intracellular uptake of HNP:** Since our previous studies were performed with PC-3 CaP cells, we assessed the uptake of our HNP in LNCaP cells to confirm uptake in these cells. Our results show that after 6 h of exposure to our Nile red encapsulating HNPs, followed by epifluorescence microscopy, intracellular uptake in LNCaP cells and LNCaP/Doc-R can be seen at similar levels to PC-3 cells as published in our earlier study (Figure 2A). This uptake is based on the intensity of red cells, which most likely is due to passive targeting by EPR effects [40].

**Intracellular Drug Concentration:** Following uptake studies, we next focused on determining the intracellular Doc concentrations of Doc in all cell lines. We performed these experiments again with LNCaP, LNCaP-Doc/R, and PC-3 cells. All cells were cultured with either free Doc (5 μg/mL) or the equivalent HNP Doc dose for 6 h. At the end of incubation, cells were lysed, and the lysates were analyzed for Doc levels. Figure 2B shows the results of these studies, and the intracellular concentrations of Doc in the LNCaP, LNCaP-Doc/R, and PC-3 cells were not statistically different and were roughly similar at 98 ± 6.7 ng/cell, 87 ± 5.8 ng/cell, and 93.6 ± 7.3 ng/cell, respectively, after 6 h of incubation. These results are consistent with our findings of similar uptake of Nile Red-labeled dye in these three cell lines, as given in Figure 2A. Additionally, PLGA, PLGA-CS, and PLGA-CS-UCNP components of HNP independently demonstrated limited toxicity at the concentrations not associated with the IC_50_ (indicated by the black dashed line (Figure 2C).

**Apoptosis Assay in resistant and naïve LNCaP cells**: Flow cytometry was performed to assess the efficacy of HNP-Doc administration in promoting apoptosis in LNCaP and LNCaP-Doc/R cells, as shown in Figure 3. The analysis revealed that treatment with 1 nM Doc induced apoptosis in both cell lines; however, the rate of apoptosis was significantly higher in LNCaP cells compared to LNCaP-Doc/R cells (26.9% vs. 18.46%; *p* < 0.01). When 1 nM HNP-encapsulated Doc was administered, the rate of apoptosis increased further, reaching 37.1% in naïve LNCaP cells. Notably, the apoptotic rate was even higher in LNCaP-Doc/R cells, rising to 45.6% (Original data generated by RWD-C100-Pro automated cell counter 100 of Annexin-V/PI stained LNCaP and LNCaP-Doc/R is attached in Appendix A). These results indicate that the HNP formulation enhances the delivery and apoptotic efficacy of Doc in Doc-resistant LNCaP cells, highlighting its potential as an effective therapeutic approach.

**Docetaxel increases cytokine production in the LNCaP cancer cell line mode:** We assessed the type and amount of cytokine released in response to either free Doc or HNP-encapsulated Doc in both the PSMA^+^ LNCaP and LNCaP-Doc/R cells. Our results, shown in Figure 4, demonstrate that treatment of LNCaP cells with free Doc results in the production of higher levels of many different cytokines, including pro-inflammatory cytokines, compared to treatment with Doc encapsulated in our HNP. The levels of several pro-inflammatory cytokines, such as C5/C5a, GCSF, GMCSF, Gro-α, I-309 IFN-γ, IL-1α, IL-1ra, IL-4, IL-6, IL-10, IL-23, I-309, and RANTES, were increased by free Doc-treated LNCaP cells, whereas their levels produced in response to HNP-encapsulated Doc were significantly lower. C5/C5a, RANTES, I-309, IL-4, IL-6, and IL-1ra were produced at >3-fold higher levels by LNCaP cells cultured with free Doc compared to HNP-encapsulated Doc (Figure 4). Similarly, C5/C5a, GCSF, increased 2-fold or greater in LNCaP cells treated with free Doc. When cytokine secretion in LNCaP cells exposed to Doc delivered via HNP was analyzed, we saw a dramatic decrease in the fold change in C5/C5a, GCCSF, RANTES, IL-1, IL-1ra, I-309, IL-6, IL-4, and IL-10.

Next, we examined the cytokine response by Doc-resistant cells in response to free Doc and HNP-encapsulated Doc. Converse to our observation with Doc-sensitive LNCaP cells, the levels of these pro-inflammatory cytokines were typically reduced in LNCaP-Doc/R cells (Figure 5), with these cells showing elevated basal levels of RANTES, IL-10, and IL-27, all of which were significantly altered following exposure to HNP-Doc (Figure 5).

**Monocyte co-culture with docetaxel-resistant LNCaP cells increased markers of macrophage differentiation:** Considering that a prior study [41], identified MIC1 as a potential contributor to Doc resistance, we subsequently examined the interaction between macrophages and CaP cells utilizing U937 cells, a promonocytic cell line capable of undergoing terminal macrophage differentiation when co-cultured with LNCaP or LNCaP-Doc/R cells in the presence or absence of Doc to assess the interaction between the macrophages and the cancer cells. Co-culture of LNCaP or LNCaP-Doc/R cells with U937 cells resulted in the adherence and differentiation of U937 cells into macrophages, as observed by their morphological changes. Analysis of conditioned media revealed that co-cultures of U937 cells with either LNCaP or LNCaP-Doc/R cells produced a significant reduction in the levels of most cytokines when compared to the media from monocultures of LNCaP or LNCaP-Doc/R cells. However, levels of interleukin-1 receptor antagonist (IL-1ra), an established marker of terminal macrophage differentiation and the tumor-associated macrophage (TAM) phenotype, exhibited a notable increase. Specifically, IL-1ra levels were elevated approximately 3.4-fold in the LNCaP/U937 co-culture system (Figure 6), and this effect was substantially amplified, reaching approximately 32-fold in the LNCaP-Doc/R/U937 co-culture system (Figure 7).

## 4. Discussion

It has been documented that early changes in circulating cytokine levels are associated with Doc resistance in CRPC patients [31,32]. Further, studies on CaP cells in vitro suggest that Doc resistance is mediated, at least in part, by the secretion of inflammatory cytokines in response to increased intracellular concentration of Doc, inducing an interaction between tumor cells and macrophages [42,43]. Despite the urgent need to understand drug resistance in CaP, research has been hampered by the limited studies on a few drug-resistant cell lines. Establishing such cell lines is essential for investigating their biological characteristics, underlying resistance mechanisms, and potential strategies to overcome resistance. In this study, we developed an in vitro Doc-resistant CaP cell line, LNCaP-Doc/R. This model was generated by intermittently exposing naïve LNCaP cells to gradually increasing concentrations of Doc over time, creating a valuable tool for studying chemotherapy resistance and analyzing its impact on cytokine secretion.

The establishment of the LNCaP-Doc/R cell line through intermittent exposure to increasing concentrations of Doc is a well-documented method for inducing drug resistance in cancer cell lines [44,45]. This approach allows for the selection of cells that can survive and proliferate despite the presence of the drug, leading to the development of a resistant phenotype [45] and achieving an in vivo resistant index for cancer cells derived from patients [46]. The observed phenotypic changes in LNCaP-Doc/R cells, such as smaller size and more irregular cell margins compared to LNCaP cells (Figure 1A), are consistent with previous studies on drug-resistant cancer cell lines [47]. These morphological alterations may be indicative of underlying changes in cellular architecture and cytoskeletal organization, which are often associated with drug resistance [48]. The use of CCK-8 to evaluate cell growth characteristics is a standard method for assessing cell viability and proliferation. Similar doubling times (Dt) for LNCaP cells and LNCaP-Doc/R cells (58.05 and 56.42 h, respectively, Figure 1B) suggest that the acquisition of Doc resistance did not significantly alter the overall growth rate of the cells in the linear phase and is in agreement with what is reported in the literature as the standard Dt for LNCaP cells [49]. This finding aligns with other studies where drug-resistant cell lines maintain comparable growth rates to their parental counterparts [50]. The dose-dependent effect of Doc on cell survival and the significantly higher IC_50_ value for LNCaP-Doc/R cells (918.2 ± 10.04 pM) compared to LNCaP cells (10.1 ± 0.1 pM) highlight the substantial resistance developed by the LNCaP-Doc/R cell line. The resistance index of ~90-fold higher in LNCaP-Doc/R cells confirms the effectiveness of the intermittent exposure method in inducing drug resistance. This result is consistent with the literature reports where resistant cell lines exhibit markedly higher IC_50_ values for the drug they were exposed to [51].

Examining gene expressions in different resistance pathway genes is a method of confirming and validating drug resistance in the studied cell line. This profiling elucidates the molecular basis of resistance, pinpointing modified genes and pathways in resistant cells. Through the analysis of these alterations, we confirm the emergence of resistance and enhance our comprehension of the underlying biological mechanisms. This method validates the resistant phenotype and identifies potential therapeutic targets, facilitating the mitigation of resistance and improving treatment effectiveness. NOTCH2 signaling is an important pathway involved in Doc resistance [52]. The first gene we analyzed was to determine if P-glycoprotein is involved in the resistance mechanism. For this, we analyzed the gene expression of the ABCB1 gene in both LNCaP and LNCaP-Doc/R cells. Our results indicate a ~10-fold increase in the expression of the ABCB1 gene (Figure 1D), aligning with the results reported by Seo et al. [53], regarding Doc resistance and involvement of the ABCB1 transporter. ABCB1, also known as P-glycoprotein, is crucial in mediating Doc resistance in CRPC [53]. As a member of the ATP-binding cassette (ABC) transporter family, ABCB1 functions as a drug efflux pump [29], thereby reducing the intracellular concentration of Doc and diminishing its therapeutic efficacy. Previous studies have demonstrated that increased ABCB1 expression is associated with acquired Doc resistance in CRPC [54,55]. Strategies to counteract this resistance, such as using specific inhibitors like elacridar or tariquidar, or employing RNA interference techniques, have been explored and underscore the role of ABCB1 in Doc resistance [56,57]. Consequently, the observed >10-fold increase in ABCB1 gene expression in LNCaP-Doc/R cells confirms the transition of LNCaP cells from naïve to a resistant cell state.

Apart from this pathway, we also evaluated the expression levels of genes from this Neurogenic locus notch homolog (NOTCH) signaling pathway, specifically NOTCH2 and hairy and enhancer of split-1 (HES1) genes. Our results showed a significant increase in the expression levels of these genes in LNCaP-Doc/R cells (Figure 1D). This increase further supports the resistance observed in LNCaP-Doc/R cells, as it is well known that elevated levels of these factors can amplify stem-like traits, suppress apoptosis, and trigger epithelial–mesenchymal transition (EMT) [58]. These mechanisms collectively contribute to the challenge of overcoming drug resistance. Consequently, targeting NOTCH2 signaling is under investigation as a therapeutic strategy to combat resistance in various cancers, including breast cancer, lung cancer, and chronic lymphocytic leukemia [59]. Research has shown that targeting the NOTCH pathway using γ-secretase inhibitors (GSIs), such as PF-03084014, can enhance the efficacy of Doc [60]. This combination has demonstrated promising results in preclinical models, reducing tumor growth and overcoming resistance by inhibiting cancer stem-like cells, angiogenesis, and survival pathways [60]. Collectively, the significant increase in the expression of NOTCH pathway genes further validates the extent of Doc resistance achieved in LNCaP-Doc/R cells. In Doc-resistant prostate cancer cells, HES1 is upregulated as part of a broader activation of the Notch signaling pathway [52], which also involves increased NOTCH2 and Hedgehog signaling. Increased expression of HES1 contributes to cancer cell survival, EMT, and stemness, which are key factors in drug resistance mechanisms. Therefore, the roughly 3-fold increase in HES1 expression observed in LNCaP-Doc/R cells confirms the resistance to Doc that has been achieved.

Another pathway that we examined for Doc resistance confirmation in this study is the mechanistic target of rapamycin complex 2 (mTORC2) pathway. In this pathway, the specific gene studied was RICTOR. Our results show a ~3.2-fold increase in the expression of this gene in LNCaP-Doc/R cells when compared to naïve LNCaP cells (Figure 1D). This increase is another confirmation of the Doc-resistant nature of our LNCaP-Doc/R cells since RICTOR, a pivotal component of the mTORC2 complex, in mediating Doc-resistance in CaP cells has garnered significant attention [61]. Enhanced expression of RICTOR is known to augment mTORC2 activity, which in turn regulates critical cellular processes such as survival, proliferation, and cytoskeletal organization [62]. This upregulation contributes to resistance mechanisms by fostering cancer cell survival and diminishing the therapeutic efficacy of Doc [62]. Recent studies have underscored the potential of targeting RICTOR or mTORC2 to mitigate this resistance [63]. For instance, the application of mTORC2 inhibitors or the silencing of RICTOR expression via RNA interference has shown promise in sensitizing prostate cancer cells to Doc in preclinical models [64]. These findings suggest that therapeutic strategies aimed at disrupting mTORC2 signaling could enhance the responsiveness to Doc, thereby improving treatment outcomes. The results presented in Figure 1 confirm that the acquisition of Doc-resistance in LNCaP-Doc/R cell line, developed through intermittent Doc exposure in naïve LNCaP cells, provides a valuable model for studying prostate cancer drug resistance. Apart from this, the results presented also identify key resistance pathways, including ABCB1-mediated drug efflux, epithelial–mesenchymal transition (EMT), NOTCH2 signaling, and mTORC2 activation. Increased expression of ABCB1, V-cadherin, vimentin, NOTCH2, HES1, and RICTOR confirms acquired resistance.

Tumor cell-derived cytokines released in response to increased intracellular concentrations of Doc are responsible for attracting monocytes and macrophages to the tumor site to induce Doc resistance. This is a great concern for continued treatment with Doc, despite the drug showing clear benefits in overall survival rates. This proinflammatory response could be potentially overcome by administering Doc encapsulated in an NF, like our HNP, theoretically protecting “Doc” from exposure to the immune system and mellowing down the cytokine release. However, results from our study with HNP-encapsulated Doc showed an increase in cellular uptake and intracellular concentration when compared to free Doc (Figure 2A,B and [28]). In theory, this could mediate a proinflammatory cytokine secretion, resulting in the induction of Doc resistance. However, since our HNP-based uptake is associated with a slow and sustained release and not a burst in Doc availability intracellularly, the release of cytokines involved in mediating Doc resistance could potentially be reduced. The PLGA-CS HNPs utilized in this investigation were synthesized by a water–oil–water emulsion technique, as elaborated in our prior publication regarding this NP [28]. The initial water-oil emulsion was created by ultrasonically combining PLGA with MC and deionized water. A chitosan (CS) wrapping was made by sonicating an initial water/oil emulsion with a secondary aqueous phase containing CS, PVA, and sodium acetate buffer. The buffer was employed to ensure the protonation of CS and to facilitate its adhesion to the negatively charged surface of the uncoated PLGA HNP. HNPs were synthesized when the MC evaporated during overnight stirring of the mixture. The nanocarriers produced in the prior study had an average size of 240 nm and a zeta potential of +40 mV, consistent with the findings of this research [28]. Incorporating Doc into the MC phase facilitated its integration, while introducing UCNPs into the first water phase enhanced their dispersion. Furthermore, the components of HNP, like PLGA, PLGA-CS, and PLGA-CS-UCNP, exhibited minimal toxicity at the concentrations corresponding to the IC_50_ (as indicated by the black dashed line, Figure 2C), which confirms that the increase in cytotoxicity is solely due to Doc as part of HNP, rather than a consequence of the toxicity of building blocks, which are part of HNP.

The observed enhancement in apoptosis (Figure 3), in Doc-resistant (LNCaP-Doc/R) cells, despite the use of the same Doc concentration (1 nM), can be attributed to the improved delivery and efficacy achieved through HNP encapsulation. Nanoparticles, including HNPs, are known to exhibit higher affinity for cancer cell membranes and may leverage specific cellular uptake pathways, resulting in superior internalization compared to free Doc [65]. Additionally, cancer cell resistance often arises due to efflux pumps like P-glycoprotein, which actively expel chemotherapeutic agents such as docetaxel from the intracellular environment [66]. Consistent with this observation, our results in Figure 1D demonstrated elevated P-glycoprotein expression in resistant cells. However, the observed increase in rate of apoptosis despite the higher expression of P-glycoprotein observed in the resistant LNCaP Cells in this study may be due to HNPs effectively shielding the drug from these efflux mechanisms, thereby circumventing a critical resistance pathway [67]. This leads to increased intracellular drug concentrations, which likely account for the significantly enhanced apoptosis observed in LNCaP-Doc/R cells. Moreover, nanoparticle encapsulation facilitates controlled release of the drug, ensuring sustained exposure to the therapeutic agent. Such prolonged drug release is particularly beneficial in resistant cells that typically require higher or sustained drug levels to achieve effective apoptosis. Additionally, nanoparticle-based delivery systems are capable of modifying drug properties, enabling them to overcome microenvironmental barriers and enhance therapeutic efficacy [68]. The observed results in an increase in the rate of apoptosis in the resistant cell line may also reflect synergistic effects arising from alterations in the pharmacokinetics or pharmacodynamics of Doc, which could create novel interactions with cellular pathways, as has been reported in other studies utilizing nanoparticle delivery systems [69,70]. Collectively, these factors highlight the advantages of the HNP-Doc formulation in effectively inducing apoptosis in Doc-resistant LNCaP-Doc/R cells, underscoring its potential to overcome the limitations of conventional drug administration methods.

To determine if this delivery of Doc via HNP encapsulation results in less cytokine release in vitro, we assessed the type and amount of cytokine released in response to either free Doc or HNP-encapsulated Doc in the conditioned media of LNCaP cells. The cytokines were quantified using the standard ELISA method. Our results in Figure 4 show that LNCaP cells treated with free Doc produced considerably higher levels of many different cytokines compared to treatment with HNP-encapsulated Doc. The levels of several pro-inflammatory cytokines, such as IFN-γ, IL-1α, RANTES, IL-1ra, IL-6, IL-23, and I-309, were increased in free Doc-treated LNCaP cells. In contrast, these pro-inflammatory cytokines were generally decreased in HNP-encapsulated Doc. Our inference from this analysis is that Doc treatment in vitro is associated with a proinflammatory response involving cytokines linked to macrophage recruitment and activation, since it is well known that some of the cytokines listed above are directly linked to this process; however, HNP-encapsulated Doc treatment leads to a lower proinflammatory response. This observation of ours confirms earlier reports on similar lines by other investigators [29,71,72]. Also, it has been reported that patients who respond poorly to chemotherapy have increased levels of IL-1ra, a marker of macrophage differentiation [31], which was also produced at higher levels by LNCaP cells cultured with Doc-as compared to LNCaP cells at a basal level in the present study. Similarly, C5/C5a, I-309, and IL-1 were increased 2-fold or greater in our study with free Doc treatment compared to HNP-encapsulated Doc. Being a monocyte chemoattractant, I-309 promotes human monocyte migration, which has a known role in developing resistance to drug treatment [73]. On the other hand, the increase in the level of IL-6 in response to free-Doc can be explained by the fact that IL-6 is a proinflammatory cytokine that inhibits cytotoxic drug-induced apoptosis, as demonstrated by Borsellino et al. [74] and Pu et al. [75], potentially through the Bcl and Stat signaling pathways [75]. IL-6 is also known to upregulate MCP1 expression, a chemotactic factor that recruits monocytes and is present at elevated levels in metastatic prostate tumor tissue [76]. In patients with CRPC undergoing Doc chemotherapy, increased baseline serum IL-6 levels are inversely associated with treatment response, time to progression, and both prostate cancer-specific and overall survival [77]. The observation that the levels of several proinflammatory cytokines in LNCaP cells exposed to Doc delivered via HNP showed a dramatic decrease agrees with the known role played by these cytokines in inflammation, and potentially due to the slow release of Doc intracellularly, as documented by us in our earlier study [28]. The increase in the level of RANTES, IL-1ra could potentially play a role in attracting immune cells like macrophages and T cells to the tumor site and creating a proinflammatory microenvironment that can support cancer cell growth and progression. The observed levels of IL-23 and I-309 (also known as CCL1) in this study on treatment with free Doc in LNCaP cells correspond with their established pro-inflammatory roles and their role in promoting tumor growth and proliferation [78,79]. Research suggests that IL-23 plays a role in promoting chemo-resistance in cancers, particularly by creating an immunosuppressive tumor microenvironment that shields cancer cells from the effects of chemotherapy; studies have shown that high levels of IL-23 can be associated with resistance to chemotherapy, especially in cancers like prostate cancer where it can be produced by myeloid suppressor cells, hindering the efficacy of ADT [80].

The basal cytokine levels secreted by LNCaP-Doc/R were distinct from those of LNCaP, with LNCaP-Doc/R exhibiting generally lower levels of cytokines (Figure 5). Intriguingly, LNCaP-Doc/R cells exhibited elevated basal levels of IL-4 and IL-10, anti-inflammatory cytokines. This interesting observation can be explained by the fact that resistant cell lines often have increased levels of anti-inflammatory cytokines compared to chemotherapy-sensitive cell lines because the resistant cells have developed mechanisms to evade the inflammatory response triggered by chemotherapy, which can otherwise lead to cell death; this can involve altered signaling pathways as well. Increased levels of anti-inflammatory cytokines can contribute to tumor immune evasion, tumor growth promotion, and tumor angiogenesis. Other resistant cells are known to decrease cytokine receptor expression, allowing them to tolerate the damaging effects of the drug and to promote survival [81]. The general decrease in the level of pro-inflammatory cytokines in LNCaP-Doc/R cells can be explained by the fact that counterbalancing of chronic activation of immune cells in rheumatoid arthritis is achieved by inhibiting the synthesis of certain pro-inflammatory cytokines and dampening immune responses in general [82].

The findings of this study highlight a marked increase in IL-4 levels in LNCaP-Doc/R cells, underscoring its potential role in driving tumor progression. This is consistent with established evidence attributing IL-4-induced tumor growth to diverse cellular mechanisms in hematopoietic, endothelial, and various tumor cell lines [53]. The tumor-promoting properties of IL-4 have been substantiated through in vitro investigations in prostate and pancreatic cancer models, where elevated IL-4 levels correlate strongly with enhanced cell proliferation and progression [54,55,56,57,58]. Clinical observations further support these findings, revealing IL-4’s role in fostering prostate tumor cell survival by impeding apoptosis [55]. In resistant tumor models spanning multiple cancer types—including colon, breast, lung, fibrosarcoma, and bladder—higher IL-4 production has been identified as a characteristic feature, distinguishing resistant cells from their non-resistant counterparts [59]. Mechanistic studies have elucidated IL-4’s interaction with its receptor, IL4-Rα, triggering upregulation of key anti-apoptotic mediators such as cFLIP, PED, FLAME-1, and Bcl-x(L), thereby facilitating tumor survival and advancement [54,60,61,62]. Moreover, IL-4 exerts a profound influence on the microenvironment of the tumor. For instance, IL-4-secreting Th2 cells amplify metastatic behavior in melanoma models [63] and drive macrophage polarization towards the M2 phenotype, which is recognized for its pro-tumorigenic properties [64]. Similarly, tumor-associated IL-4 directs CD8+ T-cells to adopt a “type 2” phenotype, impairing their anti-tumor functionality [65]. The interplay between IL-4-driven mechanisms and chronic inflammation, a recognized pathogenic factor in prostate cancer, may further compound its tumor-promoting effects [66,67,68]. Thus, two concise studies have analyzed blood IL-4 levels in CaP patients, suggesting that elevated IL-4 may correlate with the advancement of castrate resistance [83,84]. The data gathered from this study indicate elevated levels of IL-4 in the Doc-resistant cell line upon stimulation with Doc aligns with the findings of Perambakam et al. [83] and Takeshi et al. [84], and reinforces the involvement of IL-4 in the tumor growth of resistant cells. Some resistant cells can actively pump out chemotherapy drugs, reducing the amount of damage and subsequent anti-inflammatory response by increasing the secreted level of IL-4. It is not clear if this response is a cause or an effect of the pumping out of the chemotherapy drugs.

In our study, we also observed an increase in the level of IL-10 in LNCaP-Doc/R cells treated with Doc. This observation can be explained by the known effects of IL-10 on infection and tumor resistance. Our study demonstrated an increase in IL-10 levels in LNCaP-Doc/R cells treated with Doc. This finding aligns with the recognized role of IL-10 in infection and tumor resistance. As a multifunctional cytokine, IL-10 exhibits potent anti-inflammatory and immunosuppressive properties. While initially characterized as a secretion of T helper 2 cells, its production is now attributed to a diverse range of myeloid and lymphoid immune cells, underscoring its central role in both innate and adaptive immunity [71,72]. The principal function of IL-10 in therapeutic contexts is to modulate the host immune response, mitigating tissue damage and immunopathology [72,73]. Mechanistically, IL-10 suppresses pro-inflammatory cytokine production and inhibits antigen presentation in activated monocytes, macrophages, and dendritic cells, while concurrently limiting excessive T-cell activation and proliferation [71,72]. This anti-inflammatory activity is predominantly mediated through IL-10 receptor engagement, primarily expressed on monocytes and macrophages, which activates the JAK1-TYK2-STAT3 signaling cascade. The resultant STAT3-driven transcription attenuates inflammatory responses [71,72]. Further insights indicate that IL-10’s capacity to suppress pro-inflammatory cytokine expression is reliant on the inositol phosphatase SHIP1, which serves as a critical mediator [74]. Notably, the induction of SHIP1-STAT3 complex formation by IL-10 uniquely distinguishes its signaling mechanisms from other STAT3-activating cytokines, such as IL-6 [75]. These molecular interactions highlight the multifaceted and selective nature of IL-10 signaling pathways, further emphasizing its relevance in the context of tumor resistance. IL-10 facilitates tumor proliferation in CaP by inhibiting antitumor immune response via its influence on immune cells and by directly affecting CaP cells. It is also known to inhibit the antitumor immune response by suppressing the function of myeloid and T effector cells [85,86].

## 5. Conclusions

In conclusion, cytokines associated with Doc resistance in LNCaP cells have been identified through in vitro studies. The data demonstrate a correlation between the levels of macrophage-associated cytokines and Doc resistance. A major conclusion from our study is the significant decrease in inflammatory cytokines when Doc is administered in an HNP formulation, despite achieving a much higher intracellular concentration. While these findings are promising, certain limitations warrant further exploration. The exclusive focus on LNCaP and LNCaP-Doc/R cell lines may not encompass the genetic and phenotypic variability seen in patients. Additional cell lines with distinct characteristics could yield different outcomes. The use of U937 monocytes in the co-culture system does not account for macrophage diversity, including M1 and M2 polarization, which may influence responses. While HNP encapsulation of Doc appears promising, its physiological effectiveness and safety in humans remain unverified. Finally, in vitro co-culture models do not fully replicate the complexity of tumor microenvironments, where factors such as blood flow and systemic immune interactions play critical roles. Nonetheless, an alternative modality of administering Doc via HNP encapsulation could be considered for CaP treatment in the future. This approach could be further enhanced for treatment improvements by targeting the HNPs to CaP cells with a targeting moiety to their surface.

## Figures and Tables

**Figure 1 cancers-17-01758-f001:**
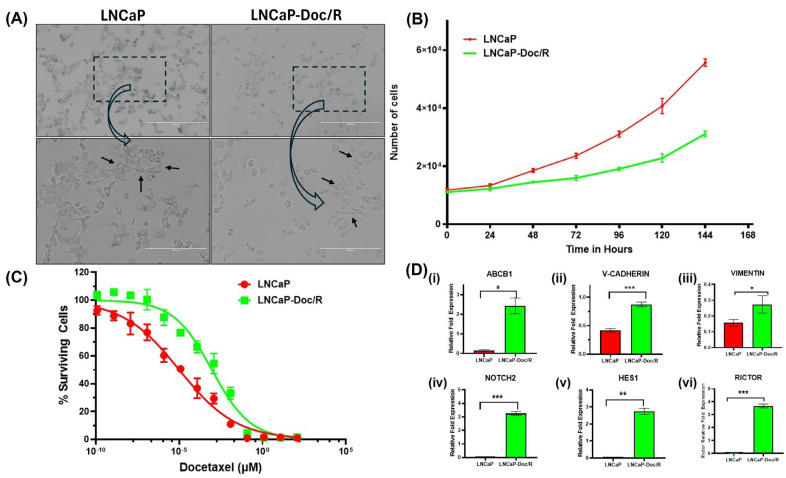
Development and characterization of Doc-resistant LNCaP-Doc/R cell line: LNCaP and LNCaP-Doc/R cells grew in a humidified atmosphere of 5% CO_2_ and 95% 0_2_ in RPMI media and treated with Doc as described in the method section. After repeated treatment with increasing amounts of Doc, cells surviving 25 nM Doc were captured and replated for imaging with an inverted microscope. (**A**) Morphologies of naïve LNCaP and LNCaP-Doc/R cells observed under an inverted microscope. The bottom panel shows a magnified view of the selected area from the upper panel, with arrows indicating typical cell morphologies. Images represent three independent experiments (n = 3). Scale bar: 200 µm. (**B**) The growth curves of LNCaP and LNCaP-Doc/R cells. The cell growth curves were plotted with culture time on the x-axis and the average number of cells per day on the y-axis. Data are expressed as the mean ± standard deviation of n = 4 experiments. The doubling time (Dt) was calculated as indicated in the methods section. (**C**) Evaluation of cytotoxicity of Doc in LNCaP and LNCaP-Doc/R cells. LNCaP, LNCaP-Doc/R, were treated with indicated doses of Doc for 72 h, and viability of surviving cells were analyzed by the 3 (4,5 dimethylthiazol 2 Yl) 2,5 diphenyltetrazolium bromide (MTT) assay using a microtiter plate reader (Bio Tek Instruments, Inc., Winooski, VT, USA) at 570 nm. IC _50_ (50% maximum inhibitory concentration) for Doc in each cell line was calculated using GraphPad Prism 10 after transformation and curve fitting. Results are the mean ± SD of four individual experiments completed in triplicate. (**D**) Relative fold expression of Doc-resistance related genes performed by quantitative PCR(Q-PCR) in LNCaP and LNCaP-Doc/R cells. (**i**) Drug transporter gene (ABCB1), (**ii**) EMT pathway gene (cadherin), (**iii**) EMT Pathway gene (vimentin), (**iv**) NOTCH and Hedgehog pathway gene (NOTCH2) (**v**) NOTCH and Hedgehog pathway gene (HES1) and (**vi**) mTOR pathway gene (RICTOR). Gene expressions are presented as relative fold expression in each cell line compared to housekeeping gene expression (β-Actin) as unity. * *p* < 0·05, ** *p* < 0·001 and *** *p* < 0.0001 compared with naïve LNCaP cells.

**Figure 2 cancers-17-01758-f002:**
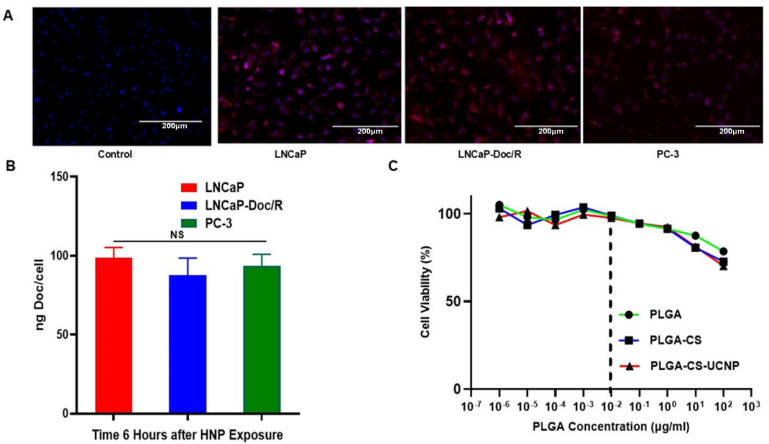
Intracellular uptake of Nile-Red encapsulating HNP by LNCaP, LNCaP-Doc/R and PC-3 cells (**A**), intracellular Doc concentration 6 h post-HNP exposure in LNCaP, LNCaP-Doc/R and PC-3 cells (**B**), IC_50_ of Free-Doc and HNP-Doc in LNCaP, LNCaP-Doc/R and PC-3 cells (**C**), and cell viability 24 h post-incubation with increasing concentrations of PLGA or PLGS-CS, or PLGA-CS-UCNP in. The dotted lines represent the maximum amount of HNP components in the formulation for the uptake studies.

**Figure 3 cancers-17-01758-f003:**
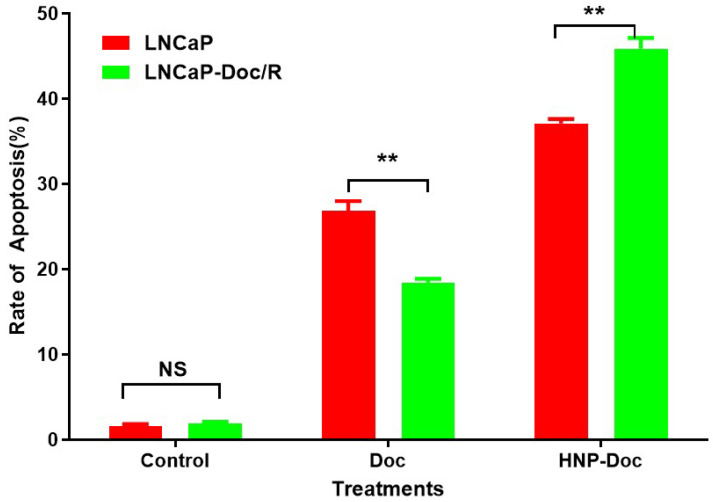
Induction of apoptosis in LNCaP and LNCaP-Doc/R cells by HNP-Doc. The percentage (%) of cell distribution in LNCaP and LNCaP-Doc/R cells 24 h after treatment with designated drugs or vehicle control, as determined by Annexin V staining. Data are presented as mean ± SD. ** *p* < 0.01 between free-Doc and HNP-Doc for each treatment, and NS—not significant. Experiments were performed at two times in triplicate.

**Figure 4 cancers-17-01758-f004:**
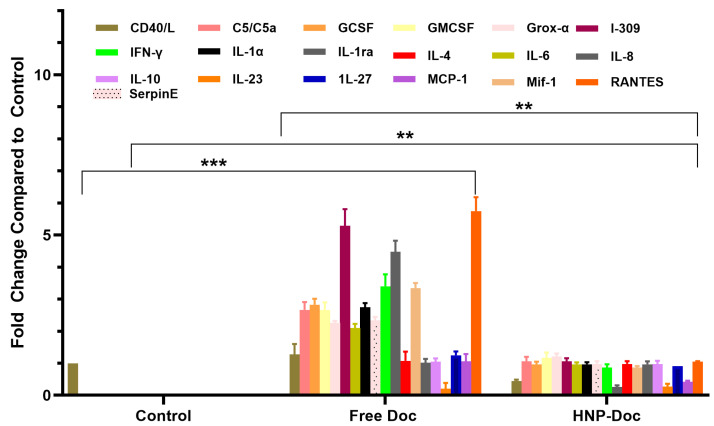
Comparison of cytokine levels in conditioned media from LNCaP Cells treated with either free Doc or HNP-encapsulated Doc. Fold change relative to LNCaP cells without any treatments. Fold change is relative to control treatment. Data are means ± S.D. from 4 experiments performed in triplicate. ** *p* < 0.01, *** *p* < 0.001.

**Figure 5 cancers-17-01758-f005:**
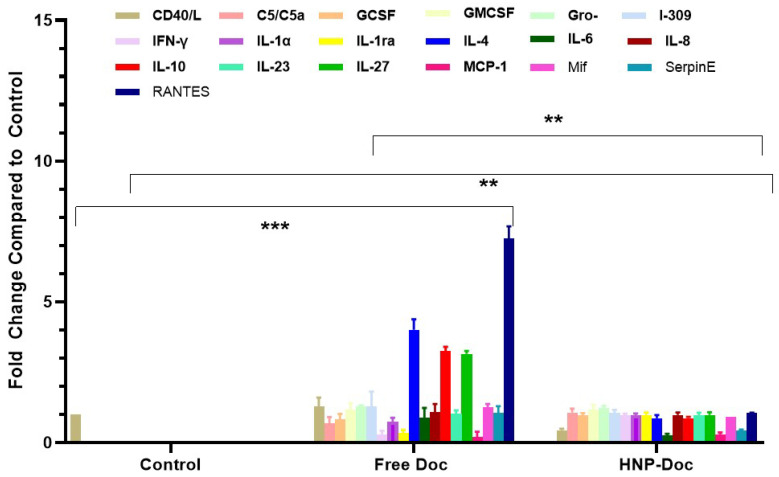
Comparison of cytokine levels in conditioned media from LNCaP-Doc/R Cells treated with either free Doc or HNP-encapsulated Doc. Fold change relative to LNCaP-Dc/R cells without any treatments. Fold change is relative to control treatment. Data are means ± S.D. from 4 experiments performed in triplicate. ** *p* < 0.01, *** *p* < 0.001.

**Figure 6 cancers-17-01758-f006:**
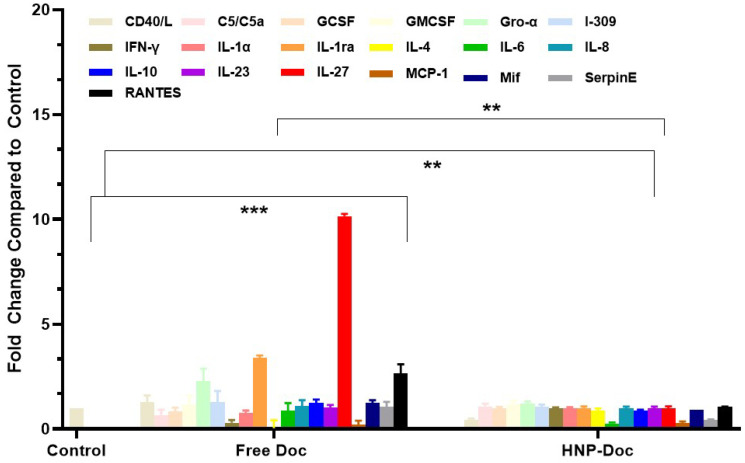
Comparison of cytokine levels in conditioned media from LNCaP Cells co-cultured with U397 cells treated with either free Doc or HNP-encapsulated Doc. Fold change relative to LNCaP and U397 cells co-culture without any treatments. Fold change is relative to control treatment. Data are means ± S.D. from 4 experiments performed in triplicate. ** *p* < 0.01, *** *p* < 0.001.

**Figure 7 cancers-17-01758-f007:**
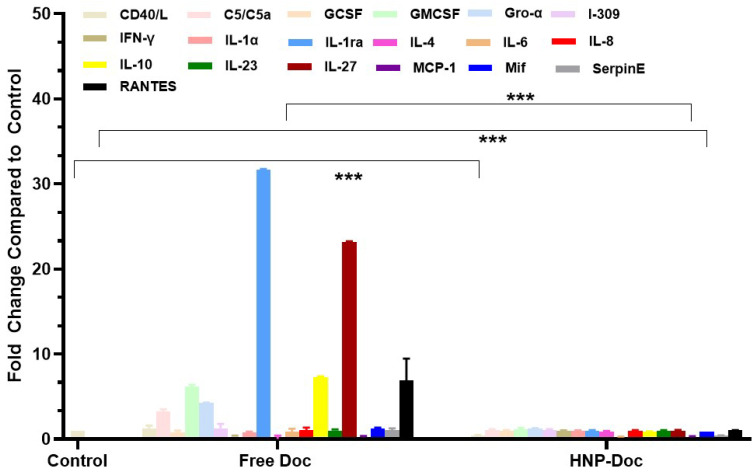
Comparison of cytokine levels in conditioned media from LNCaP-Doc/R Cells co-cultured with U397 cells treated with either free Doc or HNP-encapsulated Doc. Fold change relative to LNCaP and U397 cells co-culture without any treatments. Fold change is relative to control treatment. Data are means ± S.D. from 4 experiments performed in triplicate. *** *p* < 0.001.

## Data Availability

The data presented in this study are available on request from the corresponding author.

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
