# Peer review of "Docetaxel Administration via Novel Hierarchical Nanoparticle Reduces Proinflammatory Cytokine Levels in Prostate Cancer Cells"

_cancers, 2025, doi:10.3390/cancers17111758_

Round 1

Reviewer 1 Report

Comments and Suggestions for Authors

The manuscript was aimed to examine whether docetaxel (Doc) administration via hierarchical nanoparticle (HNP) will be beneficial against Doc-resistant prostate cancer (PC) cells and also reduce the production of proinflammatory cytokines by monocytes, thereby overcoming this resistance. 

I have the following suggestions and concerns about this manuscript: 

1) Generation of Doc-resistant PC cell line (e.g., LNCaP-Doc/R) is not confirmed and it ruins the basic concept of the manuscript. The manuscript lacks the comparative IC50 values for Doc in naive and resistant PC cells.

2) The time-period used for generation of Doc-resistant PC cell line (line 134-137) is extremely short and looks unreal. Usually its takes the couple months. 

3) What was the mechanism of Doc-resistance in LNCaP-Doc/R cell line? If the authors declare the abnormalities of Doc intracellular uptake, I strongly recommend to examine the levels of ABC-transporters reducing the intracellular concentrations of chemotherapeutic agents due to their excessive efflux. 

4) The manuscript lacks the fluorescence  images illustrating the differences in intracellulr Doc concentrations between sensitive and resistant cells. 

5) The image shown in Fig.1A is not convincing. I also recommend to perform  flow cytometry analysis to examine and compare the fluorescence intensities between these samples.

6)The manuscript lack the evidence of the enchanced efficacy of HNP-Doc against Doc-resistant PC cells. To demonstrate this, I suggest to examine the common apoptotic markers ( e.g., expression of cleaved PARP and caspase-3 by western blotting OR counting the number of Annexin V-positive cells by flow cytometry)

7) The abbreviation IL-309 (lines 239, 241, and elsewhere) is not correct. Doeas the authors mean CCL1? This should be explained. 

8) The term "hyperinflammatory cytokines" (line 15 and elsewhere) in not correct. 

Author Response

Please find a Pdf file attached with our response to Reviewers comment

Reviewer 2 Report

Comments and Suggestions for Authors

Ravikumar Aalinkeel et al., presented a research manuscript titled, “Docetaxel Administration via Novel Hierarchical Nanoparticle Reduces Hyperinflammatory Cytokine Levels in Prostate Cancer Cells”. In this work, authors aimed to modulate/reduce hyperinflammatory signals by administering Free-Docetaxel, encapsulated in a Hierarchical nanoparticle (HNP-Doc), and to evaluate its correlation between them by measuring various cytokine levels. As a result, authors suggested that the levels of several pro-inflammatory cytokines were increased in free Doc-treated prostate cancer (CaP) cells. The authors concluded that treatment involving cytokines aligns with macrophage recruitment and activation events. The introduction part covered all the basic information including recent literature reports. The methodology comprises all the experimental protocols and results presented based on the outcome of the conducted in vitro experiments. Initially, nanoparticles were synthesized and characterized instrumentally followed by cell line development and quantification. The figures are presented clearly.

Following minor revision is needed to the current manuscript:-

  1. In Results: Cite a recent reference for the section ‘Nanoparticle Synthesis…’
  2. In Conclusion, the authors need to provide the limitations of the present study.
  3. References need to be updated according to the recently published literature.
  4. IC50, 50 to be mentioned ‘subscript’
  5. Why authors selected Docetaxel (Doc) resistance in prostate cancer?
  6. Why no in vivo studies conducted to validate the in vitro results?

Author Response

Please find a Pdf file attached responding to the reviewer's comment

Reviewer 3 Report

Comments and Suggestions for Authors

Dear Author,

The article titled "Docetaxel Administration via Novel Hierarchical Nanoparticle Reduces Hyperinflammatory Cytokine Levels in Prostate Cancer Cells" presents an interesting contribution to the scientific community working on cancer therapeutics. I have some suggestions that may help improve the scientific and technical content of the manuscript and provide a better understanding of your research.

Overall, the article is well-planned and organized, demonstrating novelty and potential for translation.

  1. In the abstract, quantitative values are missing. Including these would enhance the clarity and impact of the findings.
  2. In the introduction, adding a few more references highlighting the importance of Docetaxel in the literature would strengthen the background.
  1. Pospieszna, Julia, et al. "Unmasking the deceptive nature of cancer stem cells: the role of CD133 in revealing their secrets." International Journal of Molecular Sciences24.13 (2023): 10910.
  2. Shitole, Ajinkya A., et al. "LHRH-conjugated, PEGylated, poly-lactide-co-glycolide nanocapsules for targeted delivery of combinational chemotherapeutic drugs Docetaxel and Quercetin for prostate cancer." Materials Science and Engineering: C 114 (2020): 111035.
  1. In Figure 1A, the scale bar is not clearly visible, and improving the image quality would enhance visual clarity.
  2. In Figure 1B, statistical significance is missing; including this would strengthen the presented data.
  3. In Figures 2, 3, 4, and 5, statistical significance among the groups is missing. Adding this information would improve the robustness of the results.

I hope these suggestions are helpful in improving the manuscript

Comments on the Quality of English Language

NA

Author Response

Please find a pdf file with our response attached.

Round 2

Reviewer 1 Report

Comments and Suggestions for Authors

The authors responded to my comments and suggestions. The quality of manuscript was improved. However, I strongly recommend to include the original data for new experiments that were conducted according to the recommendations. For example, FACs data for Annexing V staining is highly desirable. After this happens, the manuscript can be accepted for publication. 

Author Response

Response to Reviewer 1

Reviewer 1 comment 1: The authors responded to my comments and suggestions. The quality of manuscript was improved. However, I strongly recommend to include the original data for new experiments that were conducted according to the recommendations. For example, FACs data for Annexing V staining is highly desirable. After this happens, the manuscript can be accepted for publication.

Response: We deeply value the comments of Reviewer 1 and appreciate the acknowledgment that we responded to the comments and suggestions and are grateful to the reviewer for acknowledging its importance and impact. As recommended by the Reviewer we are attaching the Annexin V staining and the calculation of the rate of Apoptosis data. This was accomplished by use of RWD C-100-Pro Automated cell counter and not by Flow cytometry and we sincerely regret the inadvertent reference to an incorrect method in our rebuttal letter. Upon careful review, we confirm that the “Methods Section” of our manuscript correctly outlines the procedures employed in our study.
